materials science/chemical physics/ supramolecular chemistry

crystals, microstructure, banding, helical deformation, optical microscopy, birefringence

**Author for correspondence:**
Maria Raimo
e-mail: maria.raimo@cnr.it

This article has been edited by the Royal Society of Chemistry, including the commissioning, peer review process and editorial aspects up to the point of acceptance.

# An optical test to unveil twisting of birefringent crystals in spherulites

## Maria Raimo

Consiglio Nazionale delle Ricerche, Istituto per i Polimeri, Compositi e Biomateriali, Via Campi Flegrei 34, 80078 Pozzuoli, Naples, Italy

 MR, 0000-0002-2586-1996

Helical conformations and structures are frequently observed in materials. The presence of helices at points of the unit cell of a crystal, on a larger size scale in the crystalline lattice or even in the microscopic structure of crystals, affects the chemico-physical properties of a solid and, hence, also interactions with light. Here, attention has been drawn to the geometrical properties of helices produced by a hypothetical torque of a transparent crystal, and optical properties of twisted crystals easily observed by a polarizing microscope have been discussed. Radially grown spherulites are obtained by most substances crystallized from melt. The circular arrangement of elongated crystals reflects the optical behaviour of each crystal and, because of the larger dimensions of spherulites, allows investigations otherwise hardly feasible on separate crystals. According to the torsional analysis of elongated bodies and the birefringence theory, information on the existence of helically shaped crystals can be deduced, as hereinafter explained, from the microscopic appearance and birefringence pattern of spherulites. Indeed, twisting decreases the birefringence throughout an elongated crystal and, therefore, also the birefringence of spherulites formed by twisted radial crystals is reduced.

## 1. Introduction

An exhaustive analysis of the interaction of matter with light is fundamental for the achievement of technological advances and innovations. A relevant optical property of most solids is double refraction or birefringence, originated by the asymmetry of the crystalline lattice in di- and tri-metric systems, which causes the variation of the refractive index with the direction. Solids are usually highly birefringent because of the high degree of order and anisotropy of their crystalline structure. However, any oriented structure formed by asymmetric molecules, such as that of liquid crystals under particular conditions, may show birefringence. Birefringent crystals may be observed as bright

bodies in a dark field through an optical polarizing microscope equipped with a rotating stage. At a fixed goniometric position, the brightness level depends on the orientation of the crystal with respect to the propagation direction of light. The birefringent behaviour of a crystal can be described relative to one or two axes, referred to as optical axes; these axes define the only viewing directions for which a birefringent crystal is optically isotropic and, therefore, appears dark. In all other directions, transparent crystals show interference colours, varying with the thickness and the orientation of the optical axis, according to the Michel-Levy table for the identification of minerals. Maximum birefringence is shown by sections cut parallel to one of the optical axes (for uniaxial crystals, usually the $c$-crystallographic axis), whereas extinction occurs when a crystal is viewed along an optical axis, and an intermediate birefringence is appreciable for all other crystal orientations. The presence of intramolecular helices is well recognized in biological matter (typically for proteins, such as that of the tobacco mosaic virus crystals [1]) and, although less ubiquitous, helical structures may also be present in inorganic substances and minerals. For instance, the X-ray pattern and the optical behaviour of quartz evidence the presence, along the $c$ axis, of helical conformation of $SiO_4$ tetrahedrons sharing corners [2–5]. Banding such as that occasionally observed in some samples of cryptocrystalline fibres of $SiO_2$ (chalcedony) has been ascribed to alternate twisted and untwisted growth during crystallization [6], whereas twinning has been claimed to be responsible for banding in potassium chlorate crystals, where the periodical variation of the direction of the optic axis causes bright and dark bands when the crystal is viewed between crossed Nicol prisms [1].

David Brewster (who discovered both the method currently used to determine the quaquaversus polarization and the existence of biaxial crystals) represented rings in spherulites of uniaxial crystals many decades before the observation of bands in chalcedony and ascribed this optical phenomenon to rhythmic crystallization [7]. He explained extinction bands with the absence of solid or the formation of very minute crystals incapable of giving birefringence, as a consequence of a periodic 'repulsive power' keeping molecules at a distance from the growth front. Almost a century after the scientific work of Brewster, Gibbs [8] identified this repulsive power with the heat generated at the growth front during solidification and, later, Belousov [9] discovered a chemical reaction that spontaneously exhibits temporal periodicity. Meanwhile, a number of researchers came to the same conclusion as Brewster on banding [10,11]. The presence of intramolecular helices in a crystalline phase, observed in proteins as well as in synthetic polymers, is related to thermodynamic reasons. Helical morphology, instead, is usually thought to derive from screw dislocations [2,12] and to develop rapidly because of kinetic factors. Macro-helical morphology, such as that observed for graphite grown in cast iron, may arise from aggregation of platelets [13], whereas spiral growth of two rod-like phases, rotating around each other in a matrix phase, has also been hypothesized during eutectic solidification [14]. Nano- and sub-micrometre helices intermeddle with the symmetry and the chirality of crystal space groups affecting, therefore, also optical properties, particularly birefringence of solids. As theorized first by J.W. Gibbs, materials may show circular polarization, or optical rotation, in addition to linear birefringence [15,16]. This circumstance occurs not only when molecules are chiral, but also when non-chiral substances are arranged helicoidally [5]. Helices, indeed, are inherently chiral because they may be built according to opposite handedness, resulting in mirror images (referred to as enantiomorphs) that cannot be superimposed upon each other. Right-handed and left-handed helices of non-chiral molecules are isoenergetic and, therefore, should be equally probable [17,18]. Furthermore, when right-handed and left-handed helices cannot be arranged in the same crystalline lattice, crystallization should lead to a 'racemic' solid formed by 50% of each chiral crystal. Gibbs [15] also noted that a hypothetical racemic solid with right-handed helices (all parallel to the X direction) oriented normal to left-handed helices (with all axes in the Y direction) would not affect a beam of linearly polarized light entering orthogonally to the XY plane. However, the same specimen would rotate by a certain angle, in a positive or negative manner, the polarization plane if light propagates parallel to one enantiomeric type of helix. Predicting the effect of nano-helices on the optical properties is very difficult because, beyond considering the orientations of all types of helices relative to light, it is necessary to account for their relative amount, orientation and chirality. Furthermore, it is necessary to exactly identify the symmetry properties of solids among 230 Fedorov three-dimensional space groups.

Helical structures, of micrometre and sub-micrometre dimensions, in crystals might also derive from the deformation actions of external forces [19]. Twisting of an elongated body produces, indeed, co-axial helical arrangements of previously aligned points (here referred to as fibres) of the solid; such a deformation will gradually decrease from the outward to the interior of the solid. To deal with optical properties of elongated crystals, it is convenient to consider a crystal as formed by a number of

parallel fibres, all of which are made up of sequences of unit cells. Birefringence of such a crystal depends on that of all its individual fibres which, in turn, depends on the orientation of the optical axis along them. To have maximum birefringence, the optical axis of all fibres has to lay in the plane of the microscope stage for the whole fibre length. Usually, experimental observations are performed in such a way that the optical axis is oriented along the fibre axis (length-slow or positive fibres) or perpendicularly to the fibre axis (length-fast or negative fibres). Assuming that a single fibre is deformed as a cylindrical helix, the optical axis along the deformed fibre will be parallel, in the case of a length-slow fibre, or normal, in the case of a length-fast fibre, to the tangent to the helix. The optical behaviour of an elongated crystal of sub-micrometre size would be very difficult, if not impossible, to reveal by means of an optical polarizing microscope because of the limited resolving power and magnification level of the equipment. Fortunately, almost all substances solidified by cooling from melt form spherulites [7,20–23], which, under particular conditions, can reach dimensions of hundreds of micrometres or even more. Furthermore, in such a close assemblage, radial crystals are in contact to each other so as to produce the appearance of a single individual [7,22]. The circular arrangement of elongated crystals reflects the optical behaviour of each crystal and, because of the large dimensions of spherulites, allows investigations otherwise hardly feasible on separate crystals. As the aggregation of crystals reproduces, on a larger circular scale, the optical behaviour of each radial crystal, it is possible to correlate physical properties and shape of crystals within polycrystalline solids focusing on spherulites [16,20–23], which are also ideal bodies to evidence incidental twisting of crystals during growth. Indeed, the existence of helical distortions of crystals can be deduced from the microscopic appearance and the birefringence of spherulites with the appropriate size for optical observations. Here, it is shown that torsion of the crystalline structure produces a wide range of orientations of the optical axis, and the distribution of orientations affects the optical properties of the twisted crystal. The analysis of the influence of helical deformations within the crystalline structure leads to the conclusion that birefringence decreases along the whole length of the crystal. The observation of a decreased birefringence of a crystal may have different explanations, for instance, the deformation or even the change of the crystalline lattice. Furthermore, often polymer spherulites, referred to as mixed [24], show very low birefringence and lack of a Maltese cross, which indicates, in the framework of the consolidated birefringence theory, a non-uniform crystallographic orientation of the constituent radial crystals. However, the presence of twisting can be ascertained, as hereinafter described, by observing the colour pattern of a spherulite produced by a retardation plate. For the sake of simplicity, the proposed qualitative test will be here described for low birefringent solids, as polymers, which usually show interference colours from dark grey to white of the first order, according to the Newton's Table of Periodical Colours. An intermittent decrease of birefringence may be due to several reasons [7,10,15,16,23,25–27], for instance, rhythmic crystallization or the presence of molecular helices under particular conditions. Therefore, the observation of optical extinction cannot be considered as a proof of periodic crystal twisting, but only as a starting point for further and deeper investigations. The optical test here described allows us to discriminate among the main invoked causes of banding: rhythmic crystallization or molecular twisting on one side and crystal twisting on the other. Moreover, on the basis of the result of the test, one can evaluate the appropriateness of further investigations and procedures in pursuit of the true origin of banding. Notwithstanding a few authors ascribed banding in polymers to cooperative twisting of radial crystals in spherulites (i.e. to an oscillatory orientation of the optical axis with a periodic perpendicular orientation to the microscope stage causing extinction), other investigators observed that it is topologically impossible to arrange in such a way a set of ribbons [28]. Hereinafter, it is shown that torsion of real polymer crystals during their growth cannot result in a rotation of the optical axis around the geometrical axis of the crystals. Banding, indeed, mostly occurs because of pauses in the growth, often accompanied by a remarkable change in the size of constituent fibrils.

## 2. Material and methods

Bacterial poly(3-hydroxybutyrate) (PHB) grade, coded T19, $M_w = 890 \, kg \, mol^{-1}$, was supplied by Biomer (Germany) and used after drying under vacuum at 80°C. A small amount of PHB was squeezed between two glass coverslips (Linkam 3930) at 200°C onto a TechoKartell TK22 hot plate, then rapidly transferred onto a Linkam THMS 600 microscope hot stage endowed with a Linkam TMS91 temperature programmer. The sample was kept at 200°C for 2 min; afterwards, the temperature was reduced to 30°C at 20°C min$^{-1}$, in order to allow fast crystallization without significant thermal degradation. Six

specimens, with thickness approximately 30–50 μm, were crystallized. Crystallization began during cooling, after a brief induction time. Few early spherulites, often banded, nucleated in the melt faster than the most part of spherulites. Samples were kept at the final temperature of 30°C until complete crystallization. Observations were performed with a Axioscope Zeiss polarizing microscope equipped with λ/4 and λ plates (Leitz, Germany) and a JVC TK-1085 video camera coupled with the Image Pro Plus 3.0 software. Pictures were also taken with a Canon EOS 1300D camera equipped with Sigma 105 mm F2.8 EX DG Macro OS objective. To transform linear polarized light into circularly polarized light, the λ/4 plate was inserted, between the stage and the observed specimen, at 45° with respect to the polarization direction. To determine the birefringence sign of spherulites, viewed in both linearly and circularly polarized light, the λ plate was placed in between the stage and the analyser.

## 3. Theoretical background and discussion

Although radial crystals in spherulites are usually considered rectangular parallelepipeds, here the torsion of a cylindrical crystal of radius $r$, carrying a constant torque, will be discussed for simplicity. The rigorous analysis of torsion, and hence of birefringence, of more complex crystal shapes requires advanced mathematical methods, because the points of each non-circular cross-section not only rotate but also move out of the plane of the section; however, intuitive considerations lead to the conclusion that the displacement field of points of a transversal section of a non-circular prismatic bar and then the shear stress distribution are not much dissimilar to those existing in circular bars.

As twisting of a shaft produces a helical arrangement of previously aligned points, before discussing torsion it is useful to outline mathematical properties of helices. A helix, or curve of constant slope, is defined by the property that the tangent makes a constant angle with a fixed line. In particular, cylindrical helices are three-dimensional curves with zero slope. Such curves may be described by the parametric equations: $x = r\cos t$, $y = r\sin t$ and $z = kt$, where $k = p/2\pi$ is proportional to the pitch $p$ of the helix ($p$ is the distance between two consecutive points of the helix belonging to a same generatrix of the cylinder). From the definition of cylindrical helices, it comes out that the tangent at any point $P$ forms with the generatrix of the cylinder for $P$ a constant angle $\beta_h$. The angle $\beta_h$ has to differ from 0 and $\pi/2$; otherwise, a helix reduces to a circumference or a line. The relationship between the helical angle $\beta_h$, the pitch $p$ and the radius $r$ of the section of the cylinder is $\beta_h = \arctan(2\pi r/p)$. As $\beta_h$ is constant, any line, passing for a point $P$ of the helix and normal to the cylinder axis, will form with the tangent of the helix for the same point $P$ a constant angle $\alpha_h$, with $\alpha_h + \beta_h = \pi/2$. The angle $\alpha_h$ is usually referred to as the inclination of the helix. As $\tan \beta_h = \cot \alpha_h = 1/\tan \alpha_h = (2\pi r/p)$, it follows that $\tan \alpha_h = 1/\cot \alpha_h = (p/2\pi r)$, that is $\alpha_h = \arctan(p/2\pi r)$. In a Frenet frame, the unit vector $\mathbf{T}$ tangent to a cylindrical helix at a point $P$ has the following coordinates:

$$\mathbf{T} = \left( \frac{-r\sin t}{\sqrt{r^2 + k^2}}, \frac{r\cos t}{\sqrt{r^2 + k^2}}, \frac{k}{\sqrt{r^2 + k^2}} \right).$$

The vector $\mathbf{N}$ normal to the helix has the following components:

$$\mathbf{N} = \left( \frac{-r\cos t/\sqrt{r^2 + k^2}}{r/\sqrt{r^2 + k^2}}, \frac{-r\sin t/\sqrt{r^2 + k^2}}{r/\sqrt{r^2 + k^2}}, 0 \right) = (-\cos t, -\sin t, 0).$$

The vector $\mathbf{B}$ (referred to as binormal), normal to the plane of $\mathbf{N}$ and $\mathbf{T}$, has the following components:

$$\mathbf{B} = \mathbf{N} \times \mathbf{T} = \left( \begin{vmatrix} \frac{r\cos t}{\sqrt{r^2 + k^2}} & \frac{k}{\sqrt{r^2 + k^2}} \\ -\sin t & 0 \end{vmatrix}, -\begin{vmatrix} \frac{-r\sin t}{\sqrt{r^2 + k^2}} & \frac{k}{\sqrt{r^2 + k^2}} \\ -\cos t & 0 \end{vmatrix}, \begin{vmatrix} \frac{-r\sin t}{\sqrt{r^2 + k^2}} & \frac{r\cos t}{\sqrt{r^2 + k^2}} \\ -\cos t & -\sin t \end{vmatrix} \right)$$

$$= \left( \frac{k\sin t}{\sqrt{r^2 + k^2}}, -\frac{k\cos t}{\sqrt{r^2 + k^2}}, \frac{r}{\sqrt{r^2 + k^2}} \right).$$

The cosine of the angle $\beta_h$ between the vector $\mathbf{T}$ and the versor $\mathbf{V} = (0,0,1)$, which is parallel to the generatrices of the cylinder (i.e. parallel to the $z$ axis), is given by the following equation:

$$\cos \beta_h = \frac{\mathbf{T} \cdot \mathbf{V}}{\|\mathbf{T}\| \cdot \|\mathbf{V}\|} = \frac{k/\sqrt{r^2 + k^2}}{\sqrt{r^2 \sin^2 t/(r^2 + k^2) + r^2 \cos^2 t/(r^2 + k^2) + k^2/(r^2 + k^2)}} = \frac{k}{\sqrt{k^2 + r^2}}.$$

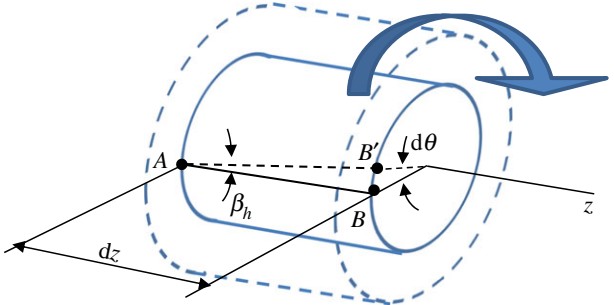

**Figure 1.** Torsion of a cylindrical bar. Each linear segment $AB$ deforms into a helix. The helical angle $\beta_h$ depends on the measure of the arc $BB'$, which increases with the distance of the fibre $AB$ from the torsion axis.

The cosine of the angle $\alpha_h = \pi/2 - \beta_h$ between the vector **B** and the vector **V** $(0,0,1)$ is as follows:

$$\cos\alpha_h = \frac{\mathbf{B}\bullet\mathbf{V}}{\|\mathbf{B}\|\bullet\|\mathbf{V}\|} = \frac{r/\sqrt{r^2+k^2}}{\sqrt{k^2\sin^2 t/(r^2+k^2) + k^2\cos^2 t/(r^2+k^2) + r^2/(r^2+k^2)}} = \frac{r}{\sqrt{k^2+r^2}}.$$

The angles $\gamma$ and $\delta$ that the tangent **T** forms with the $x$ and $y$ axes, respectively, may be obtained from the following formulae:

$$\cos\gamma = \frac{dx}{\sqrt{dx^2+dy^2+dz^2}} = \frac{-r\sin t}{\sqrt{r^2+k^2}}$$

and

$$\cos\delta = \frac{dy}{\sqrt{dx^2+dy^2+dz^2}} = \frac{r\cos t}{\sqrt{r^2+k^2}}.$$

Assuming $a = k/r$, it results:

$$\cos\gamma = \frac{-\sin t}{\sqrt{1+a^2}}$$

and

$$\cos\delta = \frac{\cos t}{\sqrt{1+a^2}}.$$

In conclusion, for a given helix, the angle $\beta_h$ (and, equivalently, the angle $\alpha_h$) is independent of $P$; also the numbers: $r/k^2+r^2$ and $k/k^2+r^2$, which represent, respectively, the curvature and the torsion of the helix, are independent of $t$ and, hence, of $P$.

For the present purpose, the torsional analysis can be limited to the determination of the relationship between the helical angle $\beta_h$ (which represents also the shear strain of a surface element of the cylinder) and the angular rotation $d\theta/dz$, which represents the angle of twist for unit length. From figure 1, it is easy to show that $BB' = rd\theta = \beta_h dz$ and, therefore:

$$\beta_h = \frac{d\theta}{dz}r.$$

As, for a prefixed torque, the helical angle $\beta_h$ is independent of $z$ but depends on $r$, fibres of the crystal at different distance from the torsion axis will form helices with different inclinations. Each of these helical pathways has a definite inclination and, consequently, will show its own orientation of the optical axis. At different distances $r$ from the geometrical axis and, therefore, at different points of any section within the crystal, the optical axis will have different orientations. For each helix at a given distance $r_i$, indeed, the optical axis in any point $P_i$ of the helix will form a constant angle with the generatrix of the cylinder passing for $P_i$. In particular, if the optical axis in the original straight fibres is oriented longitudinally, at any point $P_i$ of a helix the optical axis will have the same direction of the tangent and, therefore, will form an angle $\beta_{hi} = r_i d\theta/dz$ with the torsion axis. In other words, the change of the helical angle $\beta_h$ at different 'deep' of the twisted crystal is accompanied by a variability of the helical inclination and, therefore, produces also a variability of orientation of the optical axis in each cross-section.

According to the torsion theory, the shear stress in points of a cross-section of a cylindrical bar is proportional to the distance from the torsion axis and remains constant along each concentric circumference. For a rectangular section of a parallelepiped crystal, the stress, and hence the deformation, is maximum at the middle points of the sides and decreases, according to a nonlinear

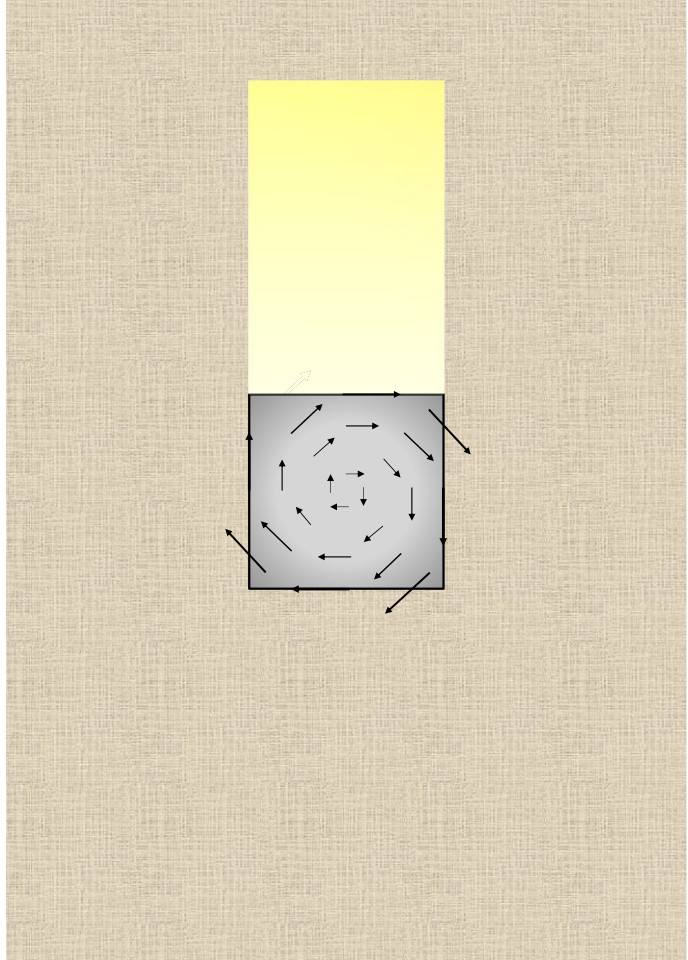

**Figure 2.** Displacement field in a square cross-section of a parallelepiped crystal submitted to torque and viewed longitudinally. In particular, when the fibres of the crystal are optically slow-length, the displacement vector in any point of the cross-section has the direction of the optical axis. In any case, a beam of light faces several orientations of the optical axis through the thickness of the crystal. Birefringence is hence reduced for the whole length of the crystal.

law, with decreasing distance from the torsion axis. In both cases, hence, the shear stress is not constant, it is directed along the tangents to circles and has the maximum value at the surface, independent of the exact geometry of the section (see, for instance, figure 2). Although the shear stress is not constant in each transversal section of an elongated crystal of a given shape, the stress distribution is the same in all sections of the crystal. And, because the stress vectors are tangential, the points of any section will suffer an average deformation equal to the average deformation of the whole points of the crystal.

The helical deformation of the crystal induced by twisting is also responsible of the rotation, around the torsion axis, of the optical indicatrix of each constituent fibre and, hence, of the birefringence change relatively to the untwisted crystal. As twisting of a shaft produces a helical arrangement of all fibres of a crystal (as shown above), the optical properties of twisted crystals can be deduced from those of co-axial hollow cylinders, each built by helices with the same inclination at equal distance from the cylinder axis and exchangeable by a simple translation along the $z$ axis.

Given a cylinder with radius $r$, there are infinite helices on it, differing for the value of $k$, that is for the values of the inclination and the pitch. However, when two among the three parameters $r$, $\alpha_h$ or $\beta_h$ and $p$ are prefixed, also the third results fixed. If the radius $r$ of the cylinder is increased, it is necessary to increase $p$ in order to keep constant $\tan \alpha_h$; whereas it is necessary to decrease $\tan \alpha_h$ to maintain $p$ unchanged. In general, it is possible to build on the surface of two cylinders with different radii two sets of helices having either different inclination or pitch. However, a cylindrical twisted crystal can be thought as built by sets of $n_1, n_2, \ldots, n_m$ helices with the same pitch $p$ wound on $m$ co-axial hollow cylinders with increasing radii $r_1, r_2, \ldots r_m$. Let us assume that any $n_i$ set of helices on the $i$th cylinder have the same inclination $\alpha_i$, that is the same value of $k = k_i$. The $n_j$ helices coiled on the $j$th cylinder

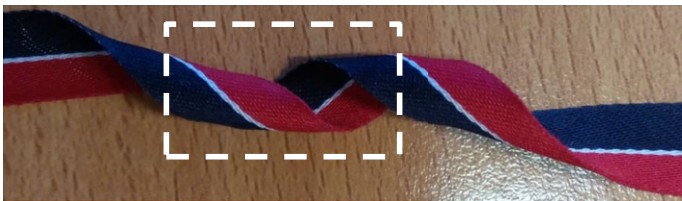

**Figure 3.** Helical winding of a ribbon-shaped crystal. If the untwisted crystal is considered as constituted of slow-length fibres, it is immediate the perception of the sinusoidal change of the projection of the tangent to each helically deformed fibre. The sinusoidal trend of the derivative of the projection of a helix entails a change in the birefringence sign every half-pitch.

with radius $r_j$, although inclined of an angle $\alpha_j \neq \alpha_i$ may be described by the same value $k$ valid for the $n_i$ helices; indeed, if $r_j \tan \alpha_j = r_i \tan \alpha_i$, it results $k_j = k_i$ and the two sets of helices $n_i$ and $n_j$ will have the same pitch, i.e. $p_i = p_j$. Torsion of a cylindrical bar will transform $n_i$ lines parallel to the torsion axis, and at equal distance $r_i$ from the axis, in helices with equal pitch $p_i$ and equal inclination $\alpha_i$. Moreover, $n_j$ lines parallel to the torsion axis and at a distance $r_j \neq r_i$ from the axis, will be deformed in helices with a different angle $\alpha_j$ but with the same pitch $p_j = p_i$ of all other helices in the bulk.

As said above, the helical deformation of the crystal structure induced by twisting is responsible for the rotation, around the torsion axis, of the optical indicatrix and, hence, of the optical axis, for each co-axial hollow cylinder within the crystal. Knowing the geometrical properties of helices, it is easier to determine, for the deformed crystalline structure, the distribution of the orientation of the optical axis within a section and the effect of helical deformations on birefringence. This distribution is shown in figure 2 for a positively birefringent crystal. Let us consider a light beam perpendicular to the geometric axis of the cylinder, assuming that the optical axis throughout the untwisted cylinder is parallel to the geometric axis. This assumption entails that the line tangent to the helix in a given point has the direction of the optical axis in the same point. If a crystal was so tiny to consist of few coplanar fibres or only one fibre, and it could be wound as a unique helix around a cylinder, practically a continuous rotation of the optical indicatrix around the helical axis of the unique fibre would be observed. This virtual circumstance would lead to the observation of successive maxima and minima of birefringence along the length of the crystal. Moreover, the projection (the white two-dimensional curve in figure 3) of the helix, viewed longitudinally, has a sinusoidal-type derivative function. Therefore, the optical axis of such a hypothetical crystal formed by one helix should fluctuate every half-pitch from the II–IV quadrants to the I–III quadrants. An elongated crystal viewed under an optical polarizing microscope with a retardation plate (generally a lamina of quartz, mica or gypsum with a definite thickness) shows a definite colour at a goniometric angle of $+45°$ and a different colour at $-45°$, when the angles are measured from the polarization plane of the light. A hypothetical single helix crystal should show bands of both colours, although reverted, either at $+45°$ and $-45°$ or, in the case of a very small pitch, a uniform colour given by the combination of the two. If, for instance, a linear fibre viewed with a retardation plate is yellow at $+45°$ and blue at $-45°$, once helically deformed it should show intermittent yellow and blue bands or a greenish colour in all sectors of the stage. However, such a narrow crystal, if really existing, would be incapable of being birefringent [7,10]. Each circular section of a real twisted cylinder contains, instead, portions of helices with angles $\beta_h$ decreasing with decreasing radius, that is with angles $\alpha_h$ increasing with $r$. If an observer could pass through the twisted cylinder perpendicularly to the torsion axis, he would face helices with different inclination at different depth and, hence, a distribution of orientation of the optical axis. Therefore, two rays perpendicular to the cylinder axis, and entering two sections, at any distance, of the distorted crystal will experience the same reduction in linear birefringence. In other words, the twisted crystal will show a uniform and lower birefringence along its length. Moreover, because along each helix within a twisted crystal, the direction of the optical axis is reverted every half-pitch, and a crystal is made of sets of co-axial helices for the whole length, the optical test settled by Brewster [7] to establish the birefringence sign of spherulites allows us also to reveal twisting. Indeed, by using a retardation plate, a hypothetical low birefringent spherulite constituted of thin, twisted radial crystals should show four luminous sectors of the same colour or alternating bands with the same colours, although exchanged, in all sectors. The colour patterns of twisted and untwisted spherulites of low birefringence solids, as polymers, are shown in figure 4. In figure 5, several PHB-banded spherulites, observed through a polarizing microscope equipped with a $\lambda$ retardation plate are shown. Spherulites were obtained with the non-isothermal crystallization

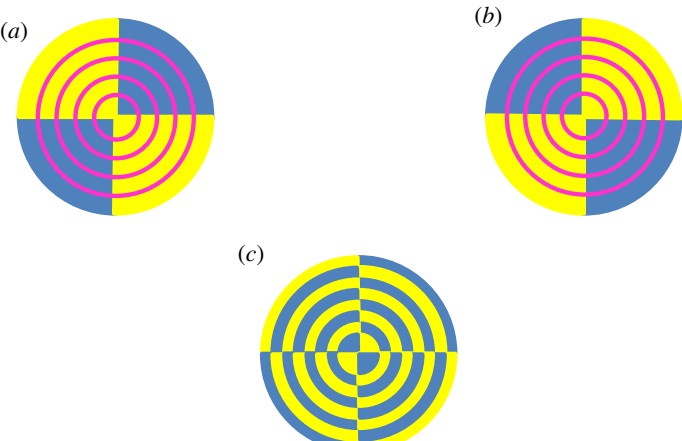

**Figure 4.** Scheme of the colours of untwisted (*a* and *b*) and twisted (*c*) spherulites observed with a λ plate. A spherulite is usually positive (*a*) or negative (*b*) according to the crystallographic orientation of radial crystals. It is worth specifying that the birefringence sign and, thus, the colours of an untwisted spherulite cannot be changed by rotating the stage. If pauses in the growth occur, non-birefringent rings within spherulites arise. A helicoidally twisted thick crystal is 'neutral' because the optical axis has not a preferred orientation and, therefore, will show one colour or, in the case of a very thin crystal, alternating colours (usually blue and yellow for weakly birefringent solids). A spherulite composed of radial twisted fibres will appear therefore coloured as the spherulite (*c*). However, if the pitch of each helical fibre is below the resolution power of the microscope, all the fibres constituting a twisted spherulite will appear greenish for the whole length.

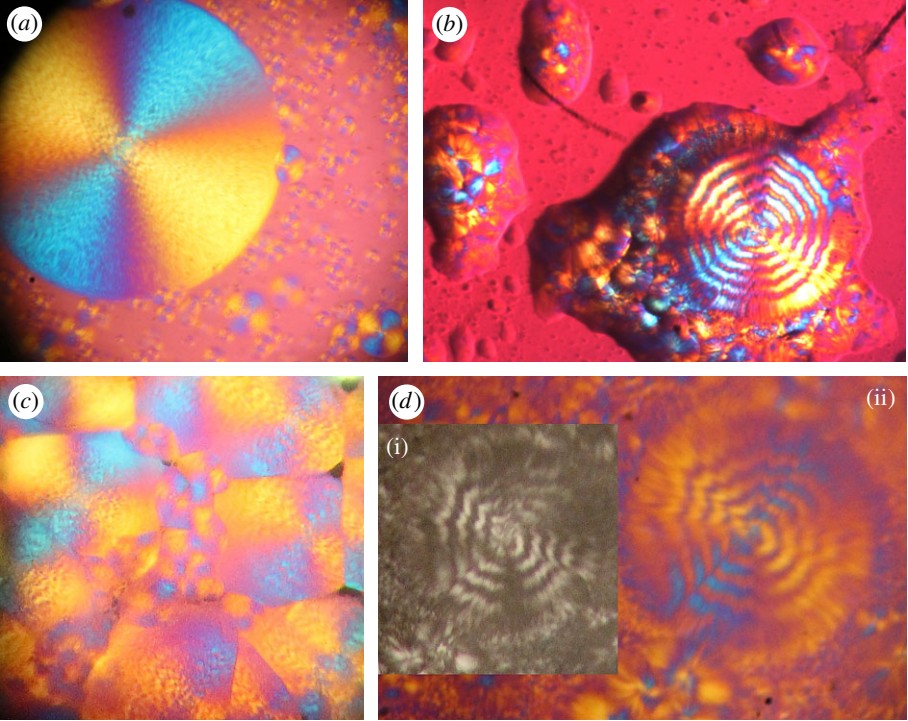

**Figure 5.** Polarized light micrographs (crossed polarizers) of three poly(3-hydroxybutyrate) specimens. Colours are produced with a first-order or full-wave retardation plate. All spherulites show crossed sectors with a prevalent yellow or blue colour onto a magenta background. (*a*) A region of a PHB specimen during non-isothermal crystallization showing a large spherulite (approx. 200 μm in diameter). Bright bands appear to be thicker than dark bands. (*b*) A spherulite (approx. 130 μm in diameter) with higher band spacing. (*c*) Positively birefringent banded spherulites in a completely crystallized PHB specimen. (*d*) A zig-zag-banded spherulite viewed without (i) and with the λ plate (ii).

described in the Material and methods section. Non-isothermal procedures were preferred to slow isothermal crystallizations, because undesired PHB degradation would favour the accumulation of lower molecular weight species at the growth front and, therefore, rhythmic crystallization over

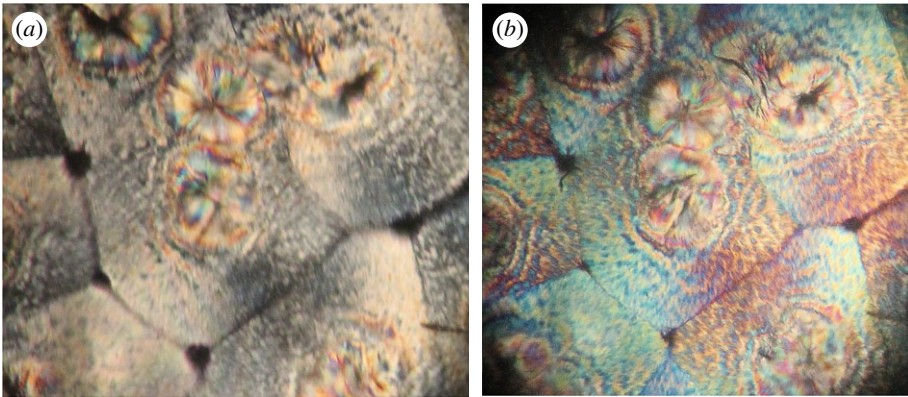

**Figure 6.** Combined optical micrographs of a PHB specimen under circularly polarized light, with (*b*) and without (*a*) a λ plate. Spherulites do not show a Maltese cross and more details are appreciable than in linearly polarized light, but their sign remains unchanged.

twisting. In figure 6, PHB specimens viewed under circularly polarized light, with and without a λ plate, are shown. Under circularly polarized light, the Maltese cross disappears and details are highlighted by a higher level of contrast. Notwithstanding the differences, spherulites in all specimens showed the same birefringence sign: because the I–III sectors are blue and the II–IV sectors are yellow, the spherulites are all conventionally positive, i.e. the value of the refractive index measured along radial directions is higher than that measured along tangential directions. Positive birefringence indicates that the optical axis of any radial fibre in a spherulite is constantly directed along the fibre axis and that, therefore, fibres cannot be helicoidally twisted. In the present case, the proposed test clearly reveals that banding cannot be due to twisting but, probably, to the oscillation of the thickness along the radii of the spherulite [19,27,29–31].

Briefly, because the optical axis is not more uniformly oriented, but has a variable direction through the thickness of a twisted cylinder, this latter will show a reduced birefringent power for the whole length, when viewed longitudinally. Moreover, the sign of the birefringence should change periodically along each helix as shown in figure 4*c*, becoming practically neutral for the whole assemblage of co-axial helices. As a consequence, a spherulite composed of radial twisted crystals should show four sectors of a sole colour.

# 4. Conclusion

Torsion of elongated birefringent crystals, with small but not null wideness and thickness, entails helical deformations of the constituent fibres in the whole volume, causing a range of orientations of the optical axis at different distances from the torsion axis, that is from the external surface to the inward of the crystal. As a consequence of geometrical properties of helices and their assembly in twisted crystals, helical distortions due to external forces during crystal growth do not cause particular optical effects but only a reduction of birefringence throughout the crystal. Dark bands alternating to birefringent bands along the length of a crystal, under a polarizing microscope, may only suggest intermittent twisting during growth, because transversal banding along crystals and concentric rings within spherulites is due to several reasons, for instance, rhythmic crystallization caused by temperature or concentration gradients at the growth front. Therefore, one cannot consider periodic extinction as a result of twisting without evidence of a cause–effect relationship between the two events. General evidence of helical twisting of an elongated crystal includes reduced birefringence and the appearance of coloured alternating bands in the presence of a retardation plate. The colours of the bands are exchanged, with a consequent inversion of the colour pattern, if the elongated crystal is rotated by 90°.

Ethics. I declare that this manuscript conforms to all ethical publication practices required by the Royal Society.

Data accessibility. The article has no data other than digital micrographs, provided with the manuscript.

Competing interests. I declare I have no competing interests.

Funding. No funding supported this research.

Acknowledgements. The author is grateful to all authors of past and current papers on the banding, which constitute the basis of this work.

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
