## [Reviewer comments · Royal Society Open Science]

Review History

RSOS-180911.R0 (Original submission)

Review form: Reviewer 1

Is the manuscript scientifically sound in its present form?

No

Are the interpretations and conclusions justified by the results?

No

Is the language acceptable?

Yes

Is it clear how to access all supporting data?

Not Applicable

Do you have any ethical concerns with this paper?

No

Have you any concerns about statistical analyses in this paper?

I do not feel qualified to assess the statistics

Recommendation?

Reject

Comments to the Author(s)

Review comments:

1. Figs. 1-3 are all drawn cartoons that are hard/impossible to be substantiated. The only exp. data are in Fig. 4. Authors did not indicate what T_c was used to produce the POM micrograph for PHB? What was the T_c in this crystallized? In addition, the specimens do not appear to be uniform. In addition to this only "big" spherulites, there are many other smaller, and non-banded, spherulites that surround this big one. From the scale as stated (130 μm for diameter, this referee calculated that the band spacing = ca. $130/12 = 11 \mu\text{m}$.

2. The photo in Fig. 3 to illustrate "helical lamella". The photo appears to be taken on a piece of cloth ribbon placed on a table and authors coiled it into a helix for taking a picture. This referee does not understand what a piece of cloth ribbon, artificially coiled and placed on table can prove anything in science? Did authors attempt to use photo of a piece of cloth, along with a crudely performed and artifact-filled POM photo, to advance the understanding of the complex banding issues that have been already debated and analyzed with all advanced instrument?

3. Ring bands and twisting have been addressed by thousands of previous papers. The whole work in this submission is superficial and filled with speculations, and unsound experimental data from a very rough and crude POM characterization (the exp. Data fig. 4, a very simple POM micrograph). If author cannot prove there is a relation between the hand-made cloth ribbon and the POM banded spherulites, this referee does not see any merits of this work?

4. Authors presented an only experimental data (POM) in this work (the other two are cartoons) - Polarized light micrograph (crossed polarizers) of a poly(3-hydroxybutyrate) banded spherulite. Not only POM characterization was crudely performed (non-uniform films, with aggregated spherulites), but also the temperature of crystallization was not indicated. Authors did not measure or indicate what "thickness" was this film? Nor did they say where is the "thickness oscillation"? How do authors explain the phenomenon that there are not only a banded spherulite but also many non-banded PHB spherulites all jammed in films of aggregated domains, where the film thickness appeared to be same for all spherulites?

5. Authors then concluded that "Banding is attributed likely to the oscillation of the thickness along the radii of the spherulite. Regardless author considered whether "twisting" is at work to be responsible for banding or not, why in the same specimens, there are tens or hundred other spherulites (neighboring this big one) that do not show bands, and only this big spherulite shows bands of spacing = 11-12 μm ?

Review form: Reviewer 2**Is the manuscript scientifically sound in its present form?**

Yes

Are the interpretations and conclusions justified by the results?

Yes

Is the language acceptable?

No

Is it clear how to access all supporting data?

Yes

Do you have any ethical concerns with this paper?

No

Have you any concerns about statistical analyses in this paper?

Yes

Recommendation?

Major revision is needed (please make suggestions in comments)

Comments to the Author(s)

Authors presented a method to test twist in crystals. Images look interesting for spherulite studies, but more statistical validation is needed.

- Minimum $n=3$ crystals should be imaged under polarization and CV values should be calculated.
- Different polarization should be tested to show birefringent crystals.
- Unique and similar crystals should be tested and their visual differences should be discussed.
- Theory and experimental match should be represented in clear graphs.

Decision letter (RSOS-180911.R0)

09-Jul-2018

Dear Dr Raimo:

Manuscript ID: RSOS-180911

Title: "A simple test to unveil twisting of birefringent crystals in spherulites"

Thank you for submitting the above manuscript to Royal Society Open Science. Your paper was sent to reviewers and their comments are included at the bottom of this letter.

In view of the concerns raised by the reviewers, the manuscript has been rejected in its current form. However, a new manuscript may be submitted which takes into consideration these comments.

Please note that resubmitting your manuscript does not guarantee eventual acceptance, and that your resubmission will be subject to peer review before a decision is made.

Your resubmitted manuscript should be submitted by 06-Jan-2019. If you are unable to submit by this date please contact the Editorial Office.

Yours sincerely,
Dr Laura Smith, MRSC
Publishing Editor, Journals
Royal Society of Chemistry,
Thomas Graham House,
Science Park, Milton Road,
Cambridge, CB4 0WF, UK

Royal Society Open Science - Chemistry Editorial Office

On behalf of the Subject Editor Professor Anthony Stace and the Associate Editor Professor Claire Carmalt

REVIEWER(S) REPORTS:

Associate Editor Comments to Author ():

RSC Associate Editor:

Comments to the Author:

(There are no comments.)

RSC Subject Editor:

Comments to the Author:

(There are no comments.)

Reviewers' Comments to Author:

Reviewer: 1

Comments to the Author(s)

Review comments:

1. Figs. 1-3 are all drawn cartoons that are hard/impossible to be substantiated. The only exp. data are in Fig. 4. Authors did not indicate what Tc was used to produce the POM micrograph for PHB? What was the Tc in this crystallized? In addition, the specimens do not appear to be uniform. In addition to this only "big" spherulites, there are many other smaller, and non-banded, spherulites that surround this big one. From the scale as stated (130 um for diameter, this referee calculated that the band spacing = ca. $130/12 = 11$ um.

2. The photo in Fig. 3 to illustrate "helical lamella". The photo appears to be taken on a piece of cloth ribbon placed on a table and authors coiled it into a helix for taking a picture. This referee does not understand what a piece of cloth ribbon, artificially coiled and placed on table can prove anything in science? Did authors attempt to use photo of a piece of cloth, along with a crudely performed and artifact-filled POM photo, to advance the understanding of the complex banding issues that have been already debated and analyzed with all advanced instrument?

3. Ring bands and twisting have been addressed by thousands of previous papers. The whole work in this submission is superficial and filled with speculations, and unsound experimental data from a very rough and crude POM characterization (the exp. Data fig. 4, a very simple POM

micrograph). If author cannot prove there is a relation between the hand-made cloth ribbon and the POM banded spherulites, this referee does not see any merits of this work?

4. Authors presented an only experimental data (POM) in this work (the other two are cartoons) - Polarized light micrograph (crossed polarizers) of a poly(3-hydroxybutyrate) banded spherulite. Not only POM characterization was crudely performed (non-uniform films, with aggregated spherulites), but also the temperature of crystallization was not indicated. Authors did not measure or indicate what "thickness" was this film? Nor did they say where is the "thickness oscillation"? How do authors explain the phenomenon that there are not only a banded spherulite but also many non-banded PHB spherulites all jammed in films of aggregated domains, where the film thickness appeared to be same for all spherulites?

5. Authors then concluded that "Banding is attributed likely to the oscillation of the thickness along the radii of the spherulite. Regardless author considered whether "twisting" is at work to be responsible for banding or not, why in the same specimens, there are tens or hundred other spherulites (neighboring this big one) that do not show bands, and only this big spherulite shows bands of spacing = 11-12 μm ?

Reviewer: 2

Comments to the Author(s)

Authors presented a method to test twist in crystals. Images look interesting for spherulite studies, but more statistical validation is needed.

- Minimum $n=3$ crystals should be imaged under polarization and CV values should be calculated.
- Different polarization should be tested to show birefringent crystals.
- Unique and similar crystals should be tested and their visual differences should be discussed.
- Theory and experimental match should be represented in clear graphs.

Author's Response to Decision Letter for (RSOS-180911.R0)

See Appendix A.

RSOS-181215.R0

Review form: Reviewer 3

Is the manuscript scientifically sound in its present form?

Yes

Are the interpretations and conclusions justified by the results?

No

Is the language acceptable?

Yes

Is it clear how to access all supporting data?

Yes

Do you have any ethical concerns with this paper?

No

Have you any concerns about statistical analyses in this paper?

No

Recommendation?

Major revision is needed (please make suggestions in comments)

Comments to the Author(s)

Raimo et al reported a test to show birefringent crystals twisting in spherulites. I suggest that authors should present complete comments from the two reviewers, and make a point-by-point reply to all those questions raised by other peer-reviewers. At current status, I can't make a decision by only checking the answers, because I don't know their questions. Thus, major revision is still requested before official publications.

Review form: Reviewer 4

Is the manuscript scientifically sound in its present form?

No

Are the interpretations and conclusions justified by the results?

No

Is the language acceptable?

Yes

Is it clear how to access all supporting data?

Not Applicable

Do you have any ethical concerns with this paper?

No

Have you any concerns about statistical analyses in this paper?

No

Recommendation?

Reject

Comments to the Author(s)

In my view the author has not done sufficient additional work to fully answer the questions posed by Referee 1. I support the view of Referee 1 that the author has not given sufficient experimental data to prove that the existence of twisting of birefringent crystals in spherulites. Figures 1 to 3 are mainly cartoons.

In my view the paper does not fully meet the criteria of the Journal i.e. the Conclusions are not sufficiently supported by the data.

In view of it, the paper is not suitable for publication in the Royal Society Open Science Journal."

Decision letter (RSOS-181215.R0)

06-Nov-2018

Dear Dr Raimo:

Title: A simple test to unveil twisting of birefringent crystals in spherulites
Manuscript ID: RSOS-181215

The editor assigned to your paper has now received comments from reviewers. We would like you to revise your paper in accordance with the referee and Subject Editor suggestions which can be found below (not including confidential reports to the Editor). Please note this decision does not guarantee eventual acceptance.

Please submit a copy of your revised paper before 29-Nov-2018. Please note that the revision deadline will expire at 00.00am on this date. If we do not hear from you within this time then it will be assumed that the paper has been withdrawn. In exceptional circumstances, extensions may be possible if agreed with the Editorial Office in advance. We do not allow multiple rounds of revision so we urge you to make every effort to fully address all of the comments at this stage. If deemed necessary by the Editors, your manuscript will be sent back to one or more of the original reviewers for assessment. If the original reviewers are not available we may invite new reviewers.

Please also include the following statements alongside the other end statements. As we cannot publish your manuscript without these end statements included, if you feel that a given heading is not relevant to your paper, please nevertheless include the heading and explicitly state that it is not relevant to your work.

• Authors' contributions

Please include an Authors' Contributions section at the end of your main text detailing the contribution of each author. All authors should have read and approved the manuscript before submission and this should be stated in the Authors' Contributions section.

The list of Authors should meet all of the following criteria; 1) substantial contributions to conception and design, or acquisition of data, or analysis and interpretation of data; 2) drafting the article or revising it critically for important intellectual content; and 3) final approval of the version to be published.

• Acknowledgements

On behalf of the Subject Editor Professor Anthony Stace and the Associate Editor Professor Claire Carmalt.

RSC Associate Editor
Comments to the Author:
(There are no comments.)

Reviewers' Comments to Author:

Reviewer: 3

Comments to the Author(s)

Raimo et al reported a test to show birefringent crystals twisting in spherulites. I suggest that authors should present complete comments from the two reviewers, and make a point-by-point reply to all those questions raised by other peer-reviewers. At current status, I can't make a decision by only checking the answers, because I don't know their questions. Thus, major revision is still requested before official publications.

Reviewer: 4

Comments to the Author(s)

In my view the author has not done sufficient additional work to fully answer the questions posed by Referee 1. I support the view of Referee 1 that the author has not given sufficient experimental data to prove that the existence of twisting of birefringent crystals in spherulites. Figures 1 to 3 are mainly cartoons.

In my view the paper does not fully meet the criteria of the Journal i.e. the Conclusions are not sufficiently supported by the data.

In view of it, the paper is not suitable for publication in the Royal Society Open Science Journal."

Author's Response to Decision Letter for (RSOS-181215.R0)

See Appendix B.

Decision letter (RSOS-181215.R1)

27-Nov-2018

Dear Dr Raimo:

Title: An optical test to unveil twisting of birefringent crystals in spherulites
Manuscript ID: RSOS-181215.R1

It is a pleasure to accept your manuscript in its current form for publication in Royal Society Open Science. The chemistry content of Royal Society Open Science is published in collaboration with the Royal Society of Chemistry.

On behalf of the Subject Editor Professor Anthony Stace and the Associate Editor Professor Claire Carmalt.

RSC Associate Editor
Comments to the Author:
The authors have addressed the reviewers concerns

Reviewer(s)' Comments to Author:

Appendix A

Dear editor, thank you for the opportunity to resubmit the manuscript. Please, be aware that I have taken into account all reviewers comments and amended the manuscript conformably, namely:

- I have specify the number of crystallized specimens and their thickness.
- I have added more images of banded spherulites (see new fig. 5), discussing the differences and similarities among banded and non-banded spherulites, observed either under linearly and circularly polarized light.
- I have illustrated in the new fig.4 the theoretical appearance of twisted and non-twisted spherulites, and explained how to make the comparison with true spherulites.
- Moreover, below I have replayed point by point to comments of the referees.

Replay to Reviewer 1

1. Figs. 1, 2 and 3 illustrate principles of the consolidated theory of torsion. I think they are necessary to draw the attention of readers on the helical geometry of fibers of a twisted body. In particular, these figures show immediately mathematical properties of helices and their connections with optical properties of helical crystals. Therefore, I have preserved them in the resubmitted manuscript.

I wish to specify that the aim of the present work is not to study the crystallization behavior of PHB, (which has been deeply and exhaustively investigated in the past, together to the variables affecting band spacing, see for instance M. Raimo *et. Al.* *J Mat Sci* (2000) 35: 523-545 or P.J. Barham *et al.*, *J Mat Sci* 1984, 19:2781-2794) nor to deny the possibility of crystal twisting, but to provide a qualitative test to reveal twisted spherulites. In order to avoid banding produced by rhythmic crystallization, which would mask or hinder twisting, thermal degradation of PHB has been minimized with the adoption of a fast non-isothermal crystallization procedure. It is well known that with this kind of procedure thermal nucleation occurs and, therefore, a wide distribution of spherulites size arises. Several spherulites, however, nucleate at lower temperatures, resulting smaller.

2. and 3. The ribbon in fig.3 provides a geometrical model of a helix, does not claim to prove anything else than mathematical properties of helices. I have nothing against advanced instruments, but I simply recognize that even the most sophisticated instrument can produce artifacts and often questionable

results. On the other hand, optical microscopy is an old but still powerful and trustworthy technique to be abandoned. Please, be aware that I do respect the work of all people but, as a researcher, I have to respect first of all facts, trying to verify if they are explainable within the framework of existing theories. That is what I have done in the present manuscript: I repeat that, although I have not been able to observe twisted spherulites, the aim of my work is not to deny crystal twisting but to make anyone able to verify twisting by means of a simple and widely available test.

4. Please note that in the resubmitted manuscript I have added more experimental details on the number of crystallized samples, their thickness, and also explained in the text the presence of spherulites of very different size. Moreover, I am not the only one to have observed intermittent crests and depressions onto the surface of banded spherulites: a huge number of authors, independently on their explanation of banding, have shown the presence of circular depressions within spherulites. Obviously, I could quote only a small number of these papers (see references 26-28 in the manuscript).
5. Small spherulites nucleated at temperatures lower than those of larger spherulites, so that pauses for heat removal from the growth front may be proportionally reduced, and may be even not appreciable at all. Please note that banding is often more evident in the core of spherulites rather than at the borders.

Replay to Reviewer 2

Please, note that I have added several more spherulite images, also under circularly polarized light, discussing differences and similarities between them. Beyond to mention the reduced birefringence of twisted crystals, I have also clearly show the theoretical appearance of helical crystals in polarized light with a full wave plate. I have also illustrated in the new Fig. 4 the qualitative test to distinguish twisted and untwisted spherulites.

Appendix B

Dear editor and Reviewers, thank you for the opportunity to revise the manuscript. Please, be aware that I have taken into account all reviewers comments and amended the manuscript conformably, namely:

- I have provided to referee to Reviewer 3 all the comments of both Reviewer 1 and Reviewer 2, that are reported at the end, together to my replay and the list of the amendments made before resubmission.
- To take in account the comments of Reviewer 4, I have better specify in the manuscript (underlined text pp.5-6) that the proposed test constitutes a preliminary examination to evaluate the convenience and the type of further investigations on the origin of banding in spherulites, and that the test is especially useful as a headlight for the choice of the most appropriate experimental procedures. I have also added two new references (24 and 28) and changed the title of the manuscript in: “An optical test to unveil twisting of birefringent crystals in spherulites”
- Here is my replay to the judgement of Reviewer 4.

I absolutely agree with Reviewer 4 that the test proposed in my manuscript does not unambiguously prove crystal twisting in spherulites and that twisting must be confirmed with further inquiries. Indeed, a reduction of birefringence is also observed in the so-called mixed spherulites with a random orientation of the optical axis. Moreover, banding is not confined to twisting structures, since it has also been connected to rhythmic crystallization in several studies and by a huge number of authors. Therefore, any explanation of the origin of banding in a spherulite has to answer the following questions: is banding due to helical structures or to rhythmic crystallization, and in the former case helices are constituted of polymer chains or, as a part of polymer literature seems to suggest, of elongated crystals? The merit of my test is that it can confirm or exclude crystal twisting, leaving in the latter case to further investigations the explanation of the origin of banding. Please, note also that, for readers' convenience, I have indicated in the text the main reasons of banding together to references that allow to prove the true cause of banding.

Reviewer 4 also complains about the scarceness of experimental work. But, I repeat, my test consists advisedly in visual observations, discussed in the framework of the well-established birefringence and torsion theories. As also pointed out by Reviewer 1, many other techniques and procedures have been utilized by several investigators, so it is unnecessary to repeat their work. Please note also that, according to the birefringence theory, the first clue of cooperative crystal twisting is the absence of a dark cross in spherulites, whereas banded spherulites usually show an inner Maltese cross under linearly polarized light.

Finally, as I already told to Reviewer 1, fig.1 and fig.2 illustrate concepts of the torsion theory, whereas fig.3 provides only a geometrical model of a helix, and they do not claim to prove anything else than physico-mathematical properties of helices. Evidences of twisting of crystals must be obviously inferred from optical appearance or other physical properties of spherulites

Yours faithfully

Maria Raimo

Comments to the Author(s)

Review comments:

1. Figs. 1-3 are all drawn cartoons that are hard/impossible to be substantiated. The only exp. data are in Fig. 4. Authors did not indicate what T_c was used to produce the POM micrograph for PHB? What was the T_c in this crystallized? In addition, the specimens do not appear to be uniform. In addition to this only "big" spherulites, there are many other smaller, and non-banded, spherulites that surround this big one. From the scale as stated (130 μm for diameter, this referee calculated that the band spacing = ca. $130/12 = 11 \mu\text{m}$).

2. The photo in Fig. 3 to illustrate "helical lamella". The photo appears to be taken on a piece of cloth ribbon placed on a table and authors coiled it into a helix for taking a picture. This referee does not understand what a piece of cloth ribbon, artificially coiled and placed on table can prove anything in science? Did authors attempt to use photo of a piece of cloth, along with a crudely performed and artifact-filled POM photo, to advance the understanding of the complex banding issues that have been already debated and analyzed with all advanced instrument?

3. Ring bands and twisting have been addressed by thousands of previous papers. The whole work in this submission is superficial and filled with speculations, and unsound experimental data from a very rough and crude POM characterization (the exp. Data fig. 4, a very simple POM micrograph). If author cannot prove there is a relation between the hand-made cloth ribbon and the POM banded spherulites, this referee does not see any merits of this work?

4. Authors presented an only experimental data (POM) in this work (the other two are cartoons) - Polarized light micrograph (crossed polarizers) of a poly(3-hydroxybutyrate) banded spherulite. Not only POM characterization was crudely performed (non-uniform films, with aggregated spherulites), but also the temperature of crystallization was not indicated. Authors did not measure or indicate what "thickness" was this film? Nor did they say where is the "thickness oscillation"? How do authors explain the phenomenon that there are not only a banded spherulite but also many non-banded PHB spherulites all jammed in films of aggregated domains, where the film thickness appeared to be same for all spherulites?

5. Authors then concluded that "Banding is attributed likely to the oscillation of the thickness along the radii of the spherulite. Regardless author considered whether "twisting" is at work to be responsible for banding or not, why in the same specimens, there are tens or hundred other spherulites (neighboring this big one) that do not show bands, and only this big spherulite shows bands of spacing = 11-12 μm ?

Reviewer: 2

Comments to the Author(s)

Authors presented a method to test twist in crystals. Images look interesting for spherulite studies, but more statistical validation is needed.

-- Minimum $n=3$ crystals should be imaged under polarization and CV values should be calculated.

-- Different polarization should be tested to show birefringent crystals.

-- Unique and similar crystals should be tested and their visual differences should be discussed.

-- Theory and experimental match should be represented in clear graphs.

Dear editor, thank you for the opportunity to resubmit the manuscript. Please, be aware that I have taken into account all reviewers comments and amended the manuscript conformably, namely:

- *I have specify the number of crystallized specimens and their thickness.*
- *I have added more images of banded spherulites (see new fig. 5), discussing the differences and similarities among banded and non-banded spherulites, observed either under linearly and circularly polarized light.*
- *I have illustrated in the new fig.4 the theoretical appearance of twisted and non-twisted spherulites, and explained how to make the comparison with true spherulites.*
- *Moreover, below I have replayed point by point to comments of the referees.*

Replay to Reviewer 1

1. Figs. 1, 2 and 3 illustrate principles of the consolidated theory of torsion. I think they are necessary to draw the attention of readers on the helical geometry of fibers of a twisted body. In particular, these figures show immediately mathematical properties of helices and their connections with optical properties of helical crystals. Therefore, I have preserved them in the resubmitted manuscript.

I wish to specify that the aim of the present work is not to study the crystallization behavior of PHB, (which has been deeply and exhaustively investigated in the past, together to the variables affecting band spacing, see for instance M. Raimo et. Al. *J Mat Sci* (2000) 35: 523-545 or P.J. Barham *et al.*, *J Mat Sci* 1984, 19:2781-2794) nor to deny the possibility of crystal twisting, but to provide a qualitative test to reveal twisted spherulites. In order to avoid banding produced by rhythmic crystallization, which would mask or hinder twisting, thermal degradation of PHB has been minimized with the adoption of a fast non-isothermal crystallization procedure. It is well known that with this kind of procedure thermal nucleation occurs and, therefore, a wide distribution of spherulites size arises. Several spherulites, however, nucleate at lower temperatures, resulting smaller.

2. and 3. The ribbon in fig.3 provides a geometrical model of a helix, does not claim to prove anything else than mathematical properties of helices. I have nothing against advanced instruments, but I simply recognize that even the most sophisticated instrument can produce artifacts and often questionable results. On the other hand, optical microscopy is an old but still powerful and trustworthy technique to be abandoned. Please, be aware that I do respect the work of all people but, as a researcher, I have to respect first of all facts, trying to verify if they are explainable within the framework of existing theories. That is what I have done in the present manuscript: I repeat that, although I have not be able to observe twisted spherulites, the aim of my work is not to deny crystal twisting but to make anyone able to verify twisting by means of a simple and widely available test.

4. Please note that in the resubmitted manuscript I have added more experimental details on the number of crystallized samples, their thickness, and also explained in the text the presence of spherulites of very different size. Moreover, I am not the only one to have observed intermittent crests and depressions onto the surface of banded spherulites: a huge number of authors, independently on their explanation of banding, have shown the presence of circular depressions within spherulites. Obviously, I could quote only a small number of these papers (see references 26-28 in the manuscript).
5. Small spherulites nucleated at temperatures lower than those of larger spherulites, so that pauses for heat removal from the growth front may be proportionally reduced, and may be even not appreciable at all. Please note that banding is often more evident in the core of spherulites rather than at the borders.

Replay to Reviewer 2

Please, note that I have added several more spherulite images, also under circularly polarized light, discussing differences and similarities between them. Beyond to mention the reduced birefringence of twisted crystals, I have also clearly show the theoretical appearance of helical crystals in polarized light with a full wave plate. I have also illustrated in the new Fig. 4 the qualitative test to distinguish twisted and untwisted spherulites.